## [Peer Review File · The EMBO Journal]

Unique territorial and compartmental organization of chromosomes in the holocentric silkworm

Jose Gil, Emily Navarette, Clio Hockens, Neil Chowdhury, Sameer Abraham, Gaetan Cornilleau, Elissa Lei, Julien Mozziconacci, Edward Banigan, Leah Rosin, Leonid Mirny, Heloise Muller, and Ines Drinnenberg

Corresponding author(s): *Ines Drinnenberg (ines.drinnenberg@curie.fr)* , *Leonid Mirny (leonid@mit.edu)*, *Heloise Muller (heloise.muller@curie.fr)*

Review Timeline:

Submission Date:	28th Aug 25
Arbitrating Reports:	9th Oct 25
Editorial Decision:	16th Oct 25
Revision Received:	17th Nov 25
Editorial Decision:	5th Dec 25
Revision Received:	22nd Dec 25
Accepted:	9th Jan 26

Editor: Hartmut Vodermaier

Transaction Report: This manuscript was transferred to The EMBO JOURNAL following peer review at another journal.

REFeree REPORTS ON REVISED MANUSCRIPT BY Drinnenberg and colleagues

REFeree 1

This revised manuscript addresses all comments previously raised by reviewers and adds new high quality data which strengthen the manuscript.

I therefore strongly believe that this manuscript is of high quality, novel and extremely robust, therefore justifying publication in [*journal name redacted*].

REFeree 2

I think the authors have done a good job revising the manuscript. Although I liked it upon the initial submission, I must say that revised paper reads even better now. In the last decade, more and more non-standard laboratory animals and plants have been explored in terms of their chromosome and nuclear organization. This manuscript fits well within that trend, and the extremely interesting phenomenon of the S-compartment deserves to be brought to the attention of the field. This work stands out as a meticulous investigation that uses several cutting-edge approaches and is performed with truly high quality, and I predict it will be highly cited by both structural and cell biologists.

I am satisfied with practically all the explanations and answers provided to my questions. In particular, the addition of super-resolution microscopy was important, as it makes the conclusions from microscopy much more convincing. However, there are still two minor points that need to be clarified before acceptance:

(1) The authors explained that they performed segmentation of chromosome signals, A-, B-, S-compartment signals using TANGO, an image analysis tool dedicated to the nuclear images. In particular, TABGO can be used for measuring distances between signal centers, between a surface and a signal center. Taken in account that the authors measured distances between center of visualized compartment to the chromosome border, why the ordinates on the graphs on revised Fig.5D still show "Fraction of the foci volume". I understood that in the initial submission, the authors used signal volumes, but in the revised version – signal centers. The graphs, however, are identical in initially submitted and revised MSs. Now I am confused: what was measured –

signal volumes contributed to a particular shell or fraction of signal centers?

(2) I also find it inappropriate to show the same exemplifying image in different figure panels. Specifically, SFig.7G (left) is the same as in Fig.6E (left); SFig.7G (right) is the same as SFig.8E (left). Reusing the same image in multiple contexts without clarification may raise concerns about the robustness of data presentation. I recommend that the authors replace the duplicated images with distinct examples from their dataset. Given that the authors state they analyzed “three biological replicates, each consisting of 50 cells”;, it should be feasible to select alternative representative images.

REFEREE 3

While the authors have acknowledged the reviewers' concerns, many of the issues raised remain only superficially addressed or have not been satisfactorily answered. For the sake of brevity, I provide a couple of examples below; however, they are representative of a broader pattern observed throughout the rebuttal and the revised manuscript.

The authors write: "Although phase separation is widely theorized to be the mechanism underlying compartmentalization [...], we found that it also cannot explain S compartments. S compartments indeed have strong intra-domain contacts and lack contacts with A and B, but crucially, they lack interactions between pairs of S domains. This is the key observation that distinguishes S compartmentalization from conventional phase separation models." They also write that they found that "the only mechanistic model that works is one with targeted loading of loop extruders to S domains".

The statements above are both incorrect. It is entirely possible for a domain to emerge via phase separation driven by a specific molecular factor distinct from those that underlie A/B compartmentalization. Such a domain –arising through phase separation rather than loop extrusion– would, by its nature, not exhibit interactions with other domains. Therefore, the authors' claim that there exists a “key observation that distinguishes S compartmentalization from conventional phase separation models” is unfounded. Equally untenable is their assertion

that only loop extrusion is capable of generating domains with “S-like” characteristics.

This type of superficial conclusion recurs throughout both the manuscript and the authors' rebuttal letter. Overall, I think that the study presents an interesting and potentially valuable contribution, particularly in providing new Hi-C data for *Bombyx mori*. However, the conclusions derived from the modelling component are untrue or substantially overstretched, which undermines the overall impact of the manuscript. As a result, the work does not meet the standards expected for publication in *[journal name redacted]*.

Authors' Response to the Referee(s) from Another Journal

Reviewer #3 (Remarks to the Author):

While the authors have acknowledged the reviewers' concerns, many of the issues raised remain only superficially addressed or have not been satisfactorily answered. For the sake of brevity, I provide a couple of examples below; however, they are representative of a broader pattern observed throughout the rebuttal and the revised manuscript.

The authors write: "Although phase separation is widely theorized to be the mechanism underlying compartmentalization [...], we found that it also cannot explain S compartments. S compartments indeed have strong intra-domain contacts and lack contacts with A and B, but crucially, they lack interactions between pairs of S domains. This is the key observation that distinguishes S compartmentalization from conventional phase separation models." They also write that they found that "the only mechanistic model that works is one with targeted loading of loop extruders to S domains".

The statements above are both incorrect. It is entirely possible for a domain to emerge via phase separation driven by a specific molecular factor distinct from those that underlie A/B compartmentalization. Such a domain — arising through phase separation rather than loop extrusion — would, by its nature, not exhibit interactions with other domains. Therefore, the authors' claim that there exists a "key observation that distinguishes S compartmentalization from conventional phase separation models" is unfounded. Equally untenable is their assertion that only loop extrusion is capable of generating domains with "S-like" characteristics.

This type of superficial conclusion recurs throughout both the manuscript and the authors' rebuttal letter. Overall, I think that the study presents an interesting and potentially valuable contribution, particularly in providing new Hi-C data for *Bombyx mori*. However, the conclusions derived from the modelling component are untrue or substantially overstretched, which undermines the overall impact of the manuscript. As a result, the work does not meet the standards expected for publication in *[journal name redacted]*.

The reviewer seems confused about one or more points about our modeling approach and phase separation, and they must have overlooked the actual results of our modeling. Statements criticized by the reviewer are strongly supported by modeling presented in the manuscript, not "superficial conclusions":

- 1) The reviewer states: "It is entirely possible for a domain to emerge via phase separation driven by a specific molecular factor distinct from those that underlie A/B compartmentalization."

We indeed tested this model by simulating attractive homotypic interactions between S sites, i.e. - as suggested by the reviewer - "distinct from those that underlie A/B compartmentalization". We found that while such a model isolates S from A and B, it promotes contacts between distinct S domains (Figure S5B). Thus, while a *single* S domain could indeed segregate from A/B regions, multiple S domains would interact with each other – **something the reviewer has overlooked in our results and did not consider in their argument**. Such enrichment of interactions between S domains **conflicts with the experimental observation**, as in Hi-C, S domains are depleted in contacts with each other (Figs. 2A, 3A). Based on this contradiction between the phase separation mechanism and the experimental Hi-C data we dismiss phase separation as a mechanism of S-domain formation.

- 2) The reviewer claims: "Such a domain — arising through phase separation rather than loop extrusion — would, by its nature, not exhibit interactions with other domains."

As described in the previous point, **phase separation naturally leads to aggregation of many S domains in conflict with the experimental observation** that S domains are also secluded from each other. Such aggregation and mixing of otherwise "excluded" domains in simple phase separation has been previously considered theoretically (for example, Semenov and Rubinstein, *Macromolecules* 2002; Erdel and Rippe, *Biophys J* 2018, among others) and observed in our simulations (Fig. S5B).

- 3) The reviewer suggests that other mechanisms are "capable of generating domains with S-like characteristics."

In their previous review, the reviewer specifically proposed phase separation and TAD-like loop extrusion could either explain S compartments. In the revised manuscript, **we extensively and systematically tested these models** (Figs. 3, 4, S5, S6). They simply **cannot explain the experimental observations** of S compartments. Of the tested models, only preferential loading of loop extruders to S reliably produces the patterns consistent with Hi-C for *B. mori* (Fig. 3E, 4H, 4I, S5D). The proposed mechanism of S-domain formation by loop extrusion is also consistent with multiple pieces of evidence from the experimental Hi-C maps (Fig. 2J-K), which independently indicate increased loop extrusion activity in S. In the absence of viable alternative mechanisms, we consider the evidence in support of the model with targeted loading to S domains to be conclusive.

In summary, **we have fully addressed the reviewer's previous critiques in a systematic manner.** The assertions made by the reviewer in the most recent review are either confused or simply incorrect. We are open to suggestions to tune specific text to improve the clarity of our presentation. However, we respectfully disagree with the reviewer's broad assertion that our conclusions are "superficial," "untrue," or "overstretched"; our conclusions are well supported by experimental evidence, polymer modeling, and established physical theory.

Dear Ines,

I have now heard back from two arbitrating referees of our journal, whom I had invited to comment on the issues that original referee 3 had continued to raise - as their technical nature had made it challenging for me to simply overrule them based on your response letter, even though it seemed well-argued. As you will see from the feedback copied below, our advisors both feel that conclusions regarding the mechanism underlying "S-compartment" formation may need to be stated less assertively. However, arbitrator 1 also brings a more general reservation regarding the uniqueness of the "S-compartment" compared to known chromosomal domains; and I admit I have to agree with the referee's sentiment regarding coining of new definitions unless there is a highly compelling rationale. We would nevertheless still consider the study and its findings interesting and valuable, and in principle suitable for EMBO Journal publication; but it seems that moderating several claims would be warranted (and possibly trying the FISH experiment suggested by arbitrator 1). I would therefore like to give you a chance to consider the arbitrating reviews below together with you collaborators, after which we could maybe find a chance to discuss this further via Zoom call next week?

With kind regards,
Hartmut

Arbitrating Referee #1 (Report for Author)

Gil et al. have analyzed genome organization in in the silkworm *Bombyx mori* which comprises holocentric chromosomes. Using Hi-C, chromatin tracing and polymer modeling they describe the organism's genome organization with a strong focus on a novel compartment S, which is distinct from the commonly observed A and B compartments. Based on modeling the authors propose that compartment S is generated by high local loop-extrusion activity and low extrusion activity in the rest of the genome.

The value of this study is the description of genome organization in *B. mori*. The data are of interest because they represent an organism with holocentric chromosomes and they

nicely complement the current literature on genome organization in species across evolution.

There are two major weaknesses in the study.

First, although the study shows that the *B. mori* genome exhibits some unique features, it is not clear that these should be defined as a separate compartment for several reasons.

To start with, the definition of this new "compartment" was not fully clear. I am not convinced that the new compartment is not essentially TADs which interact less prominently with each other than in other organisms. As the authors point out, *B. mori* has less inter-chromosomal interactions and the "S compartment" may simply reflect a general reduced interaction in the organism, without constituting a distinct compartment. The features that are listed as making the S compartment distinct from A compartment TADs are not very strong (lower gene density etc.).

The authors define compartment S based on Hi-C data, but do not validate the compartment sufficiently by independent means. An important question to ask is how the proposed compartment S relates to the centromeric regions by chromatin tracing using FISH. Furthermore, the authors show FISH data of compartment S relative to A and B. But the key experiment would be to look at multiple S-compartments on the same chromosome and show that they do not interact as predicted by their modeling.

Finally, considering the radically different nature of chromosomes in *B. mori*, it is not surprising that its genome organization shows different features. It does not seem prudent to set a precedent whereby every unique feature of a genome in an organism leads to the definition of a new compartment/domain etc. Designating a novel compartment S, vis-à-vis compartments A and B, would make sense if compartment S was also found in other organisms as is the case for A and B. This is not shown and as such the data describes a feature of *B. mori* but not a general novel genome feature that should be marked by the definition of a new compartment. This field is already muddled by many "fuzzy" definitions and adding one more seems a step in the wrong direction.

A second limitation of the study is that the mechanism by which compartment S forms is derived purely from simulations and remains speculative, but is not tested experimentally. I found the arguments regarding the lack of a role for phase separation in formation of compartment S plausible but not compelling in the absence of experimental data. Although a simulation may be able to explain a particular chromatin feature, that

does not mean that the mechanism at hand is indeed the correct one. Furthermore, no predictions from the model are experimentally tested. The various statements regarding the certainty of the proposed mechanism (for example, in the abstract: "...can only be explained by... ") are overly strong and should be removed and the mechanism presented as consistent with the model, but speculative. Experimental validation of the mechanism for generation of these chromatin features is needed to be compelling and to justify this structure a new compartment.

Overall, the data show that the genome organization is distinct in *B. mori* from other organisms, which is not surprising considering the drastically different nature of its chromosomes. The definition of a new compartment in the absence of any indication of relevance beyond *B. mori* and of a clearly demonstrated mechanism for its formation, however, seems premature, not well justified, and has the potential to confuse the field.

Arbitrating Referee #2 (Report for Author)

I have read the manuscript, and the responses of the authors to the comments of Reviewer #3. In my view, the authors' responses are adequate, and I did not agree with all of concerns raised by this reviewer.

This is a very interesting study; the discovery of compartment S is exciting and the modelling makes a good case for a role of loop extrusion rather than phase separation. The modelling analyses are really extensive and done by one of the world-leading labs. Of course, the loop extrusion explanation should eventually be further tested by wet-lab perturbation experiments (cohesin/NIPBL depletion, etc), but this is certainly beyond the scope of the current manuscript.

It would, however, be good to tone down some of the overly confident wording of the abstract:

"Polymer simulations *show* that this contact pattern can only be explained by high loop-extrusion activity within compartment S, combined with low extrusion elsewhere through the genome. *This unique, targeted extrusion*..."

-- *show* is a bit too strong; "indicate" or "suggest" would be better, considering that wet-lab perturbations were not done. Computational models are powerful, but not definite proof.

-- *This unique, targeted extrusion* also sounds like it is an established fact. Perhaps it

could be reworded into something like "The targeted extrusion that we propose..."

It may be good to check the remainder of the text for similar strong wording, and it would be good to emphasise in the Discussion that future wet-lab perturbation experiments will be needed to further study the role of loop extrusion.

Without reservation I recommend publication in EMBO J.

Dr. Ines Anna Drinnenberg
Institut Curie
UMR3664
Paris
France

16th Oct 2025

Re: EMBOJ-2025-122047-T
Unique territorial and compartmental organization of chromosomes in the holocentric silkworm

Dear Ines,

Thank you again for transferring your previously reviewed manuscript to The EMBO Journal, and for sharing your thoughts on the comments of our two arbitrating referees, and how they could best be addressed. As we discussed, it will be helpful to include results from the ongoing analyses of additional Oligopaint FISH data, and otherwise important to state specific conclusions more carefully, and to more explicitly explain the rationale and reasoning underlying certain interpretations and implications. With these further revisions, we would be happy to proceed with publication of the final version of the study in The EMBO Journal.

When preparing your revised manuscript, please also make sure to modify the article format according to the instructions given in our online Guide to Authors, and in particular consider the following editorial points:

GENERAL:

- Please download and complete our author checklist (link provided below).
- Please upload the manuscript text (including figure legends) as an editable text file, and all figures (main & expanded view figures that will be typeset) without legends as individual image files with sufficient resolution/quality for production.
- Please provide suggestions for a short 'blurb' text prefacing and summing up the conceptual aspect of the study in two sentences (max. 250 characters), followed by 3-5 one-sentence 'bullet points' with brief factual statements of key results of the paper; they will form the basis of an editor-written 'Synopsis' accompanying the online version of the article. Please also upload a synopsis image, which can be used as a "visual title" for the synopsis section of your paper. The image should be in PNG or JPG format, and please make sure that it remains in the modest dimensions of (exactly) 550 pixels wide and 300-600 pixels high.

TEXT:

- Please adjust the order as well as the headers of the different manuscript sections: Title page with complete author information, Abstract, Keywords, Introduction, Results, Discussion, Methods, Data Availability, Acknowledgements, Disclosure and Competing Interests Statement, References, Main Figure Legends, Tables, Expanded Figure Legends.
- Please include full author names (rather than just first name initials) on the title page.
- On the abstract page of the manuscript, please include 4-5 general keyword terms to enhance searchability.
- Please include a Disclosure and competing interests statement (next to the Acknowledgment section) - for details, see <https://www.embopress.org/competing-interests>
- Please include a dedicated "Data Availability" section at the end of the Material and Methods (suggested wording: "The [structural coordinates | microarray | mass spectrometry] data from this publication have been deposited to the [name of the database] database [URL] and assigned the identifier [accession | permalink | hashtag].") This section should also mention any deposited code. Note that only datasets newly generated during the course of this study should be listed here.
- Please adjust the format of the reference list and of the in-text citations according to EMBO Journal format (alphabetical order, author name et al + year, first up to 10 authors should be listed, followed by 'et al'), also check that all citations are complete with complete volume and page numbers (DOI info only in case of advance publications that do not yet have volume/page numbers). Finally, please adjust the format for citation of bioRxiv preprints: The citation in the text should be: "(preprint: NAME1 et al, YEAR)"; and in the reference list: "NAME1, NAME2, ... (YEAR) article title. BIORXIV doi: XXX"

- Please make sure to state all relevant funding information both in the manuscript's acknowledgement section and equally in our submission system.

- All Materials and Methods need to be described in the main text using our 'Structured Methods' format. The Methods section should include a (separately uploaded) Reagents and Tools Table (downloadable at <https://www.embopress.org/page/journal/14693178/authorguide#structuredmethods>) listing key reagents, experimental models, software and relevant equipment, and including their sources and relevant identifiers; and a part describing the methods (ideally using a step-by-step protocol format to facilitate adoption of the methodologies across labs).

DATA:

- Please refer to our author guide (www.embopress.org/page/journal/14602075/authorguide#expandedview) regarding "supplementary figures", and consider re-organizing the current figures and supplemental figures. We do not have a specific limit for the number of figures, and we can accommodate a hierarchy of main figure - expanded view figures (which will be typeset and directly accessible in the online version) - Appendix Figures.

- Similarly, tables come as main in-text tables (with numbering), EV tables uploaded individually (and in the case of XLSX spreadsheets, containing legend information in a separate Legends tab), and Appendix tables.

- The "Appendix" PDF, which needs to be headed with a title & table of contents page, is in essence a collection of Appendix Figures, Appendix Tables, any Appendix Methods etc, and Appendix references if needed. For information on how to call out these different items in the text, please again refer to the above-linked section of our author guidelines.

- Some of the more complex supplementary tables/datasets should be converted into Expanded View Dataset - which could also combine several related tabs, and which again should have a separate Legends tab.

- Finally, you shall also receive a separate message from our Source Data curation team, with instructions on how to prepare and upload relevant image and numerical raw data.

I am returning the manuscript to you now for the final revisions, with the hyperlink below allowing you to upload all files once you are ready. Should you need additional guidance/feedback regarding this final adjustments, please do not hesitate to contact us directly. Thank you again for the opportunity to consider this work for The EMBO Journal, and I look forward to receiving your final version!

With kind regards,

Hartmut

9) To facilitate reproducibility and cross-laboratory adoption of methodologies, please structure the Materials & Methods section as outlined in our guide to authors, including a completed Reagents and Tools Table that can be downloaded from our author guidelines as well (<https://www.embopress.org/page/journal/14602075/authorguide#structuredmethods>).

10) Digital image enhancement is acceptable practice, as long as it accurately represents the original data and conforms to community standards. If a figure has been subjected to significant electronic manipulation, this must be clearly noted in the figure legend and/or the 'Materials and Methods' section. The editors reserve the right to request original versions of figures and the original images that were used to assemble the figure. Finally, we generally encourage uploading of numerical as well as gel/blot image source data; for details see: embopress.org/page/journal/14602075/authorguide#sourcedata

In the interest of ensuring the conceptual advance provided by the work, we recommend submitting a revision within 3 months (14th Jan 2026). Please discuss the revision progress ahead of this time with the editor if you require more time to complete the revisions. Use the link below to submit your revision:

Link Not Available

Referee #1:

Gil et al. have analyzed genome organization in the silkworm *Bombyx mori* which comprises holocentric chromosomes. Using Hi-C, chromatin tracing and polymer modeling they describe the organism's genome organization with a strong focus on a novel compartment S, which is distinct from the commonly observed A and B compartments. Based on modeling the authors propose that compartment S is generated by high local loop-extrusion activity and low extrusion activity in the rest of the genome.

The value of this study is the description of genome organization in *B. mori*. The data are of interest because they represent an organism with holocentric chromosomes and they nicely complement the current literature on genome organization in species across evolution.

There are two major weaknesses in the study.

First, although the study shows that the *B. mori* genome exhibits some unique features, it is not clear that these should be defined as a separate compartment for several reasons.

To start with, the definition of this new "compartment" was not fully clear. I am not convinced that the new compartment is not essentially TADs which interact less prominently with each other than in other organisms. As the authors point out, *B. mori* has

less inter-chromosomal interactions and the "S compartment" may simply reflect a general reduced interaction in the organism, without constituting a distinct compartment. The features that are listed as making the S compartment distinct from A compartment TADs are not very strong (lower gene density etc.).

The authors define compartment S based on Hi-C data, but do not validate the compartment sufficiently by independent means. An important question to ask is how the proposed compartment S relates to the centromeric regions by chromatin tracing using FISH. Furthermore, the authors show FISH data of compartment S relative to A and B. But the key experiment would be to look at multiple S-compartments on the same chromosome and show that they do not interact as predicted by their modeling.

Finally, considering the radically different nature of chromosomes in *B. mori*, it is not surprising that its genome organization shows different features. It does not seem prudent to set a precedent whereby every unique feature of a genome in an organism leads to the definition of a new compartment/domain etc. Designating a novel compartment S, vis-à-vis compartments A and B, would make sense if compartment S was also found in other organisms as is the case for A and B. This is not shown and as such the data describes a feature of *B. mori* but not a general novel genome feature that should be marked by the definition of a new compartment. This field is already muddled by many "fuzzy" definitions and adding one more seems a step in the wrong direction.

A second limitation of the study is that the mechanism by which compartment S forms is derived purely from simulations and remains speculative, but is not tested experimentally. I found the arguments regarding the lack of a role for phase separation in formation of compartment S plausible but not compelling in the absence of experimental data. Although a simulation may be able to explain a particular chromatin feature, that does not mean that the mechanism at hand is indeed the correct one. Furthermore, no predictions from the model are experimentally tested. The various statements regarding the certainty of the proposed mechanism (for example, in the abstract: "...can only be explained by...") are overly strong and should be removed and the mechanism presented as consistent with the model, but speculative. Experimental validation of the mechanism for generation of these chromatin features is needed to be compelling and to justify this structure a new compartment.

Overall, the data show that the genome organization is distinct in *B. mori* from other organisms, which is not surprising considering the drastically different nature of its chromosomes. The definition of a new compartment in the absence of any indication of relevance beyond *B. mori* and of a clearly demonstrated mechanism for its formation, however, seems premature, not well justified, and has the potential to confuse the field.

Referee #2:

I have read the manuscript, and the responses of the authors to the comments of Reviewer #3. In my view, the authors' responses are adequate, and I did not agree with all of concerns raised by this reviewer.

This is a very interesting study; the discovery of compartment S is exciting and the modelling makes a good case for a role of loop extrusion rather than phase separation. The modelling analyses are really extensive and done by one of the world-leading labs. Of course, the loop extrusion explanation should eventually be further tested by wet-lab perturbation experiments (cohesin/NIPBL depletion, etc), but this is certainly beyond the scope of the current manuscript.

It would, however, be good to tone down some of the overly confident wording of the abstract:

"Polymer simulations *show* that this contact pattern can only be explained by high loop-extrusion activity within compartment S, combined with low extrusion elsewhere through the genome. *This unique, targeted extrusion*..."

-- *show* is a bit too strong; "indicate" or "suggest" would be better, considering that wet-lab perturbations were not done.

Computational models are powerful, but not definite proof.

-- *This unique, targeted extrusion* also sounds like it is an established fact. Perhaps it could be reworded into something like "The targeted extrusion that we propose..."

It may be good to check the remainder of the text for similar strong wording, and it would be good to emphasise in the Discussion that future wet-lab perturbation experiments will be needed to further study the role of loop extrusion.

Without reservation I recommend publication in EMBO J.

Arbitrating Referee #1 (Report for Author)

Gil et al. have analyzed genome organization in the silkworm *Bombyx mori* which comprises holocentric chromosomes. Using Hi-C, chromatin tracing and polymer modeling they describe the organism's genome organization with a strong focus on a novel compartment S, which is distinct from the commonly observed A and B compartments. Based on modeling the authors propose that compartment S is generated by high local loop-extrusion activity and low extrusion activity in the rest of the genome.

The value of this study is the description of genome organization in *B. mori*. The data are of interest because they represent an organism with holocentric chromosomes and they nicely complement the current literature on genome organization in species across evolution.

We thank the reviewer for their positive feedback and their comments that we are addressing below.

There are two major weaknesses in the study.

First, although the study shows that the *B. mori* genome exhibits some unique features, it is not clear that these should be defined as a separate compartment for several reasons.

To start with, the definition of this new "compartment" was not fully clear. I am not convinced that the new compartment is not essentially TADs which interact less prominently with each other than in other organisms. As the authors point out, *B. mori* has less inter-chromosomal interactions and the "S compartment" may simply reflect a general reduced interaction in the organism, **without constituting a distinct compartment**. The *features that are listed as making the S compartment distinct* from A compartment TADs are not very strong (lower gene density etc.).

Compartments have always been detected by clustering of regions with distinct contact patterns (Lieberman-Aiden et al., 2009 (PMID: 19815776), Spracklin et al., 2023 (PMID: 36550219)). Associations of different compartment types with different combinations of histone modifications were used to classify them further, compare across species, and assign functional roles. Consistently, we detected S automatically by clustering of genomic loci by their contact patterns -- a method similar to that used by Spracklin et al. (PMID: 36550219) or Xiang and Ma (PMID: 31699985). Thus, by this formal definition, S constitutes a distinct compartment. This distinct contact pattern for S is absolutely evident from a glance at the Hi-C map. We would also like to point out that S cannot be assigned as a subcompartment of either A or B because contacts between S and A or S and B are equally low (Figure 2H).

Moreover, S is not a TAD, because different TADs don't show distinct patterns of contacts at distances >1Mb. S compartments differ not only in their "near-diagonal" (<1Mb) presentation but most evidently in their long-range interactions. Such long-range intra-chromosomal interactions are abundant in *B. mori*, for A and B, but distinctly not for S -- thus doesn't reflect lack of contacts at large separations.

We have now extended our explanation in the results section to describe how compartments including S, A and B were called by clustering their whole-chromosome contact profiles, following approaches used in previous studies to define compartments in

other organisms including humans. To the same section, we have also added an explanation how S, like other compartments, is distinguished by its global contact pattern, rather than by local contact patterns, which characterize other local structures including TADs, fountains and jets.

Finally, to specifically address the reviewer's comment on how S is distinct from A, we have now included another panel B in Figure EV3 showing that unlike A, S domains are depleted of the active histone mark H3K36me3.

Figure 1: Scatterplots showing the relative ChIP-Seq enrichment profiles of H3K36me3 and H4K20me1 per A, B and S domain separately, or combined in the same plot.

The authors define compartment S based on Hi-C data, but do not validate the compartment sufficiently by independent means. An important question to ask is how the proposed compartment S relates to the centromeric regions by chromatin tracing using FISH.

We agree with the reviewer that the relationship between compartment S and the holocentric organisation of *B. mori* chromosomes is an interesting topic of investigation. In contrast to other organisms, studies in *B. mori* derived cell lines showed that centromeric regions are not associated with a specific sequence (satellites,..) that can easily be traced using FISH. Furthermore, their locations are not fixed and instead centromere formation occurs in a manner that is recessive to active chromatin (PMID: 33125865) - an epigenetic landscape that is more consistent with compartment B than S. Given these previous findings, we consider it unlikely that there is a strong correlation between centromeric and compartment S domains.

Furthermore, the authors show FISH data of compartment S relative to A and B. But the key experiment would be to look at multiple S-compartments on the same chromosome and show that they do not interact as predicted by their modeling.

We appreciate the reviewer's suggestion. We have now added new data to the manuscript quantifying the distance between two S domains in comparison to two B and two A domains on the same chromosome. These data show that the distance between the two S domains was significantly larger than between the two A or B domains, as expected given the

peripheral localization of S domains on chromosome territories and depletion of Hi-C contacts between different S domains (Figure S7G-J).

Figure 2: (A-C) 3-color Oligopaint FISH labeling two domains from the same compartment (G = A compartment, H = B compartment, I = S compartment) and the Chr04 CT using wide-field microscopy. Bottom right of just one compartment paints next to 3D reconstructions of CTs using TANGO. (D) Center-center distance measurements for compartments A, B and S measured using TANGO. *** $p=0.0002$, * $p=0.01$. unpaired t-test with Welch's correction. Each dot represents measurements from a single CT. Data shown are from two biological replicates where $n>100$ nuclei were analyzed.

Finally, considering the radically different nature of chromosomes in *B. mori*, it is not surprising that its genome organization shows different features. It does not seem prudent to set a precedent whereby every unique feature of a genome in an organism leads to the definition of a new compartment/domain etc. Designating a novel compartment S, vis-à-vis compartments A and B, would make sense if compartment S was also found in other organisms as is the case for A and B. This is not shown and as such the data describes a feature of *B. mori* but not a general novel genome feature that should be marked by the definition of a new compartment. This field is already muddled by many "fuzzy" definitions and adding one more seems a step in the wrong direction.

We agree with the reviewer, that we would like to see similar analysis done for other organisms, including non-model organisms, something that we plan to focus on in coming years.

As mentioned in the response to the previous comment, the definition of the S compartment follows the same formal principles used to define other compartments by clustering of regions with distinct long-range whole-chromosome contact patterns. In contrast, other *local* patterns (e.g. stripes, jets, fountains etc) have not been identified using such a procedure, they do not show distinct patterns of long-range $>1\text{Mb}$ interactions, and have not been referred to as compartments.

A second limitation of the study is the that the mechanism by which compartment S forms is derived purely from simulations and remains speculative, but is not tested experimentally. I found the arguments regarding the lack of a role for phase separation in formation of compartment S plausible but not compelling in the absence of experimental data. Although a simulation may be able to explain a particular chromatin feature, that does not mean that the mechanism at hand is indeed the correct one. Furthermore, no predictions from the model are experimentally tested. The various statements regarding the certainty of the proposed mechanism (for example, in the abstract: "...can only be explained by... ") are overly strong and should be removed and the mechanism presented as consistent with the model, but

speculative. Experimental validation of the mechanism for generation of these chromatin features is needed to be compelling and to justify this structure a new compartment.

We agree with the reviewer that the central prediction of our modeling, i.e. formation of S compartment by SMC-mediated loop extrusion has not been directly tested by depletion of such factors. Presence of S-specific loop extrusion remains a valuable and exciting prediction made by our study. We have revised the text of the abstract, introduction and discussion to clarify that the mechanism is supported by, and consistent with, our models and to highlight these future avenues of investigation.

At the same time, our prediction is already supported by parallel observations of additional extrusion-specific features in Hi-C such as TADs, dots, and stripes -- seen in S. (Note that these features have not been used neither to define S, nor to fit the model). Another prediction of the model -- i.e. peripheral chromosomal location of S compartments -- has been tested by microscopy (see Fig 5).

Overall, the data show that the genome organization is distinct in *B. mori* from other organisms, which is not surprising considering the drastically different nature of its chromosomes. The definition of a new compartment in the absence of any indication of relevance beyond *B. mori* and of a clearly demonstrated mechanism for its formation, however, seems premature, not well justified, and has the potential to confuse the field.

We respectfully disagree because S here has been identified and defined through the same procedure that is used to define compartments -- clustering by long-range interactions. This makes S-compartments distinct from many local structures and patterns found in Hi-C maps in recent years.

Arbitrating Referee #2 (Report for Author)

I have read the manuscript, and the responses of the authors to the comments of Reviewer #3. In my view, the authors' responses are adequate, and I did not agree with all of concerns raised by this reviewer.

This is a very interesting study; the discovery of compartment S is exciting and the modelling makes a good case for a role of loop extrusion rather than phase separation. The modelling analyses are really extensive and done by one of the world-leading labs. Of course, the loop extrusion explanation should eventually be further tested by wet-lab perturbation experiments (cohesin/NIPBL depletion, etc), but this is certainly beyond the scope of the current manuscript.

It would, however, be good to tone down some of the overly confident wording of the abstract:

"Polymer simulations **show** that this contact pattern can only be explained by high loop-extrusion activity within compartment S, combined with low extrusion elsewhere through the genome. **This unique, targeted extrusion**..."

-- **show** is a bit too strong; "indicate" or "suggest" would be better, considering that wet-lab perturbations were not done. Computational models are powerful, but not definite proof.

-- **This unique, targeted extrusion** also sounds like it is an established fact. Perhaps it could be reworded into something like "The targeted extrusion that we propose..."

It may be good to check the remainder of the text for similar strong wording, and it would be good to emphasise in the Discussion that future wet-lab perturbation experiments will be needed to further study the role of loop extrusion.

Without reservation I recommend publication in EMBO J.

We thank the reviewer for their positive feedback. We now also toned the manuscript according to the reviewer's suggestions. We agree with the reviewer that S-localized extrusion is an exciting prediction of our study that is yet to be tested directly experimentally. We now mention this in our discussion.

5th Dec 2025

Re: EMBOJ-2025-122047R

Unique territorial and compartmental organization of chromosomes in the holocentric silkworm

Dear Ines,

Thank you for submitting your re-revised manuscript to our editorial office. I have now looked through the latest version and your responses to the two arbitrating reviewers, and I am happy to say find the study now suitable for EMBO Journal publication. Prior to final acceptance, there are only a few important editorial issues that I need to ask you for:

- Please carefully go through the reference list, which still contains several entries lacking full citation information such as volume or page/eLocator numbers; or in some cases (Buitinck et al) even journal information.

- For preprint citations: Please change in-text reference format according to "preprint: Miller et al, 2025"; and in the reference list, name of the platform plus DOI + preprint label - e.g. "bioRxiv doi: 1234/002.dfv123 [PREPRINT]"

- In the Data Availability section, please include a specific URL linking to the database in which the deposited data can be accessed.

- Please double-check to make sure to all relevant funding information in the manuscript is congruent with the info entered into our submission system. Currently missing in the submission system are:
Fondation Schlumberger (FSER202202015420); National Institutes of Health (NICHD; 1K99HD104851); missing grant number DK015602 for National Institute of Diabetes and Digestive and Kidney Diseases, National Institutes of Health (NIDDK)

- Please include Table 1 in the main text file, between the main and the Expanded View figure legends.

- With regards to EV Tables and EV Datasets:

In Dataset EV1, please include a separate "legends" tab in the spreadsheet containing the name and information on the included datasets.

Only Tables EV1, 2, 12 and 13 conform to table format, they should be renamed to "Table EV1-4" (make sure to also update all in-text references to them).

The other EV tables should be converted into additional EV Datasets - updating source file names, titles, legends and in-text references accordingly. Importantly, they should all include a header (or legend).

- Finally, during routine pre-acceptance checks, our data editors have raised the following queries regarding figures, data, and legends; I would appreciate if you briefly answered to them in the cover letter of your final submission, and made the requested text modifications with changes/additions highlighted via the "Track changes" option, to facilitate our final checking"

1. Please define the annotated p values ****/****/**/* as well as provide the exact p-values for the same in the legend of figure EV 7d as appropriate.

2. Please note that the exact p values are not provided in the legends of figures 1b; 6c, f; EV 3e; EV 8h

3. Please indicate the statistical test used for data analysis in the legends of figures 5e; EV 7c, d

4. Please note that the box plots need to be defined in terms of minima, maxima, centre, bounds of box and whiskers, and percentile in the legends of figures 2g

5. Please note that information related to n is missing in the legends of figures 1b; 5b, e; EV 3e; EV 7a, c; EV 8d

6. Please note that the error bars are not defined in the legends of figures 5b, e; EV 7a, c, j

7. Please note that the scale bar is missing for figures EV 7k, l

8. Please note that the scale bar needs to be defined for figures 6e; EV 7b; EV 8e

I am returning the manuscript to you for a final round of minor revision, solely to allow you to make these modifications and upload the revised files. Once we will have received them, we should be ready to swiftly proceed with formal acceptance and production of the manuscript.

With kind regards,

Hartmut

Hartmut Vodermaier, PhD
Senior Editor, The EMBO Journal

*** PLEASE NOTE: All revised manuscript are subject to initial checks for completeness and adherence to our formatting guidelines. Revisions may be returned to the authors and delayed in their editorial re-evaluation if they fail to comply to the following requirements. As a first step please read our guidelines for revised submissions:
<https://link.springer.com/journal/44318/submission-guidelines#cms-Revised-submissions>

1) Every manuscript requires a Data Availability section (even if only stating that no deposited datasets are included). Primary datasets or computer code produced in the current study have to be deposited in appropriate public repositories prior to resubmission, and reviewer access details provided in case that public access is not yet allowed.

4) Each main and each Expanded View (EV) figure should be uploaded as individual production-quality files (preferably in .eps, .tif, .jpg formats). For suggestions on figure preparation/layout, please refer to our Figure Preparation Guidelines:
<https://media.springernature.com/original/springer-cms/rest/v1/content/27825798/data/v1>

6) Please complete our Author Checklist, and make sure that information entered into the checklist is also reflected in the manuscript; the checklist will be available to readers as part of the Review Process File.

8) Please note that supplementary information at EMBO Press has been superseded by the 'Expanded View' for inclusion of additional figures, tables, movies or datasets; with up to five EV Figures being typeset and directly accessible in the HTML version of the article.

9) To facilitate reproducibility and cross-laboratory adoption of methodologies, please structure the Materials & Methods section as outlined in our guide to authors, including a completed Reagents and Tools Table.

10) Digital image enhancement is acceptable practice, as long as it accurately represents the original data and conforms to community standards. If a figure has been subjected to significant electronic manipulation, this must be clearly noted in the figure legend and/or the 'Materials and Methods' section. The editors reserve the right to request original versions of figures and the original images that were used to assemble the figure. Finally, we generally encourage uploading of numerical as well as gel/blot image source data.

In the interest of ensuring the conceptual advance provided by the work, we recommend submitting a revision within 3 months (5th Mar 2026). Please discuss the revision progress ahead of this time with the editor if you require more time to complete the revisions. Use the link below to submit your revision:

Link Not Available

The authors have addressed all minor editorial requests.

Dr. Ines Anna Drinnenberg
Institut Curie
UMR3664
Paris
France

9th Jan 2026

Re: EMBOJ-2025-122047R1
Unique territorial and compartmental organization of chromosomes in the holocentric silkworm

Dear Ines,

Thank you for submitting your final revised manuscript for our consideration. I am pleased to inform you that we have now accepted it for publication in The EMBO Journal.

You may qualify for financial assistance for your publication charges - either via a Springer Nature fully open access agreement or an EMBO initiative. Check your eligibility: <https://link.springer.com/journal/44318/how-to-publish-with-us>

With kind regards,

Hartmut

Please note that it is The EMBO Journal policy for the transcript of the editorial process (containing referee reports and your response letters) to be published as an online supplement to each paper. If you should prefer removal of any referee-only figures included in the point-by-point response(s), e.g. because they may still be used for future publication or because they have been reproduced from published work by others, please do let us know immediately via response email.

More information is available here: <https://link.springer.com/partners/embo-press/editorial-policies#Peer%20review>